# Fucoidan Supplementation Improves Antioxidant Capacity via Regulating the *Keap1/Nrf2* Signaling Pathway and Mitochondrial Function in Low-Weaning Weight Piglets

**DOI:** 10.3390/antiox13040407

**Published:** 2024-03-28

**Authors:** Chenggang Yin, Qingyue Bi, Wenning Chen, Chengwei Wang, Bianca Castiglioni, Yanpin Li, Wenjuan Sun, Yu Pi, Valentino Bontempo, Xilong Li, Xianren Jiang

**Affiliations:** 1Key Laboratory of Feed Biotechnology of Ministry of Agriculture and Rural Affairs, Institute of Feed Research, Chinese Academy of Agricultural Sciences, Beijing 100081, China; 82101231287@caas.cn (C.Y.); biqingyue981106@126.com (Q.B.); 82101215363@caas.cn (W.C.); liyanpin@caas.cn (Y.L.); sunwenjuan@caas.cn (W.S.); lixilong@caas.cn (X.L.); 2College of Agriculture, Yanbian University, Yanji 133000, China; 3College of Life Science, Jiangxi Science and Technology Normal University, Nanchang 330013, China; 4Institute of Agricultural Biology and Biotechnology (IBBA-CNR), Via Einstein, 26900 Lodi, Italy; casti@ibba.cnr.it; 5Department of Veterinary Medicine and Animal Science (DIVAS), University of Milan, 26900 Lodi, Italy; valentino.bontempo@unimi.it

**Keywords:** fucoidan, lipopolysaccharide, liver, oxidative stress, low-weaning weight piglets

## Abstract

Fucoidan (FC) is known for its antioxidant properties, but it has unclear effects and mechanisms on weaned piglets. Two experiments were conducted to determine the optimal FC dosage in piglet diets and its protective effect against lipopolysaccharide (LPS)-induced oxidative stress. In experiment one, 24 low weight weaned piglets were randomly assigned to four dietary treatments: a basal diet (FC 0), or a diet supplemented with 150 (FC 150), 300 (FC 300), or 600 mg/kg FC (FC 600). In experiment two, 72 low-weaning weight piglets were randomly allocated into four treatments: a basal diet (CON), or 300 mg/kg of fucoidan added to a basal diet challenged with LPS (100 µg LPS/kg body weight) or not. The results showed that FC treatments increased the G:F ratio, and dietary FC 300 reduced the diarrhea incidence and increased the plasma IGF-1 concentrations. In addition, FC 300 and FC 600 supplementation increased the plasma SOD activity and reduced the plasma MDA concentration. LPS challenge triggered a strong systemic redox imbalance and mitochondrial dysfunction. However, dietary FC (300 mg/kg) supplementation increased the activity of antioxidant enzymes, including SOD, decreased the MDA concentration in the plasma and liver, down-regulated *Keap1* gene expression, and up-regulated *Nrf2*, *CAT*, *MFN2*, *SDHA*, and *UQCRB* gene expression in the liver. These results indicated that dietary fucoidan (300 mg/kg) supplementation improved the growth performance and antioxidant capacity of low-weaning weight piglets, which might be attributed to the modulation of the *Keap1/Nrf2* signaling pathway and the mitochondrial function in the liver.

## 1. Introduction

In current intensive farming, the employment of high-yielding sows coupled with early weaning strategies has become a prevalent practice, helping to enhance sow productivity and bolster the economic yield of farms [1,2]. However, a consequence of this practice is the surge in the number of low-weaning weight piglets. The digestive system of these low-weaning weight piglets is not fully mature, the growth rate is relatively slow, the resistance to external stimuli is poor, and the diarrhea incidence and mortality rate are often high [3,4,5]. Early weaning would trigger a series of complex physiological and psychological reactions in piglets, including changing eating habits and adapting to the new social environment, which may lead to their physiological and immune function abnormalities, causing serious oxidative stress [6,7]. Oxidative stress is due to the overproduction of reactive oxygen species (ROS) that exceeds the processing capacity of the antioxidant system, so maintaining the redox balance within cells is crucial for maintaining intestinal homeostasis [8,9]. In addition, the early weaning may lead to oxidative stress and mitochondrial dysfunction in the liver, which in turn affects its redox status and function [10]. It has been found that weaning stress can cause oxidative stress and oxidative damage in the liver of piglets, activate the *MAPK* pathway, increase the apoptosis of liver cells, and affect liver function [11,12]. A disturbance in the balance of oxidation reduction can interfere with the biological processes of mitochondria, leading to mitochondrial dysfunction [13]. Although mitochondria play a crucial role in energy production, they are also one of the main organelles that produce ROS [14]. Mitochondrial dysfunction can lead to excessive ROS production and affect liver function [15].

At present, research focusing on enhancing the body’s antioxidant capacity through the inclusion of functional additives in the diet is gaining increasing attention. Previous studies in our laboratory have found that functional feed additives can enhance the antioxidant capacity of piglets [7,8], improve mitochondrial function [9], and promote the growth of piglets. Recent research in our laboratory has revealed that functional feed additives can improve the antioxidant capacity of weaned piglets by activating the *Nrf2* signaling pathway and optimize their mitochondrial function, thus effectively reducing the oxidative damage of paraquat to the liver [16]. The *Nrf2* signaling pathway plays a central regulatory role in the oxidative stress response in vivo, and it controls the expression of a variety of genes or enzymes related to antioxidants [17]. In the non-stressed state, *Keap1* forms ubiquitin E3 ligase complexes with CULLIN3 (CUL3), resulting in the polyubiquitination of *Nrf2*, which is then rapidly degraded by the proteasome system. However, under electrophilic or ROS stress, the active cysteine residue of *Keap1* is directly modified, which reduces the ubiquitin E3 ligase activity of the *Keap1–CUL3* complex, thereby stabilizing *Nrf2* [18,19]. Therefore, the regulation of *Nrf2* by *Keap1* plays a crucial role in cell resistance to oxidative stress and maintenance of cell homeostasis.

Fucoidan is a natural polysaccharide compound derived from kelp and brown algae, first identified in 1913 by the Swedish scientist Professor Kylin, who ultimately christened it “Fucoidan” [20]. As a substance with significant antioxidant properties, Fucoidan has been the subject of extensive research in recent years. Previous in vitro and in vivo studies found that fucoidan could improve cell viability and protect cells from oxidative damage induced by 2,2′-Azobis (2-methylpropionamidine) dihydrochloride (AAPH) by clearing intracellular ROS and inhibiting cell apoptosis. Fucoidan improves the survival rate of zebrafish by eliminating ROS, inhibiting lipid peroxidation, inhibiting cell death, and thus inhibiting oxidative stress [21]. A further study demonstrated that the oral intake of fucoidan decreased the concentrations of reactive oxygen species (ROS) and malondialdehyde (MDA) in mouse serum, while simultaneously boosting the activity of glutathione peroxidase (GSH-Px) and superoxide dismutase (SOD), increasing the production of adenosine triphosphate (ATP), and restoring the concentrations of mitochondrial respiratory chain complexes in cardiac tissue, thereby reducing oxidative stress and preventing mitochondrial functional damage [22]. Furthermore, fucoidan demonstrated the ability to decrease lipid peroxide (LPO) and MDA concentrations in the liver. However, it is noteworthy that higher doses of fucoidan (2000 mg/kg) may potentially trigger inflammation and metabolic disorders [23]. Drawing from the findings of prior studies, we found that fucoidan has potential as a new antioxidant, but there is a need to pay attention to the dosage in use.

To our knowledge, the literature offers limited insights into the impact of fucoidan on weaned piglets, particularly with regard to the effects of the dosage of fucoidan and antioxidant capacity of weaned piglets. In this study, we focused on piglets with lower weaning weights, who are more prone to oxidative stress. We designed two experiments with the aim of preliminarily determining the optimal dosage of fucoidan and its effect on the plasma antioxidant status of low-weaning weight piglets. Subsequently, an oxidative stress model was established by stimulating low-weaning weight piglets with lipopolysaccharide (LPS). This model was utilized to further investigate the mechanisms by which fucoidan regulates oxidative damage in low-weaning weight piglets.

## 2. Materials and Methods

### 2.1. Animal Ethics Approval

The trial was conducted from July to August 2022 and September to October 2023 at the Tianpeng Experimental Farm located in Langfang, Hebei province. The animal protocol in this study was approved by the Animal Care and Use Committee of the Institute of Feed Research of the Chinese Academy of Agricultural Sciences (IFR-CAAS20221010 for experiment one and IFR-CAAS20230825 for experiment two).

### 2.2. Animals and Treatment

Experiment one: In reference to the previous selection scheme [5,24], this study selected 72 healthy weaned piglets (Duroc × Landrace × Yorkshire), with an average body weight (BW, 6.62 ± 0.13 kg) and age (25 ± 1 days). These weaned piglets were categorized into high, medium, and low weight groups based on their body weight. A total of 24 weaned piglets with low-weaning body weight (BW, 5.81 ± 0.05 kg) were randomly divided into 4 treatment groups with 6 replicates per group and 1 piglet per replicate. The dietary treatments were as follows: basal diet without FC (FC 0), FC 0 + 150 mg/kg FC (FC 150), FC 0 + 300 mg/kg FC (FC 300), or FC 0 + 600 mg/kg FC (FC 600). The experiment lasted for 21 days and plasma samples were collected (Figure 1A). The Fucoidan used in this study was purchased from Zhenlu Biotechnology Co., Ltd. (Xi’an, China), purity ≥ 98% and derived from kelp.

Experiment two: According to the selection protocols of experiment one, this study selected 72 healthy low weight weaned piglets (BW, 6.01 ± 0.32 kg) from 216 healthy weaned piglets (Duroc × Landrace × Yorkshire) with an average body weight (BW, 8.14 ± 0.15 kg) and age (28 ± 1 day). They were randomly allocated into 4 treatments with 6 replicates per treatment and 3 piglets per pen: a basal diet (CON) or 300 mg/kg of fucoidan added to a basal diet (FC) challenged with LPS or not. On day 21, one piglet was selected from each pen. Piglets in the challenged groups were intraperitoneally injected with 1 mL of LPS (*Escherichia coli* O55:B5, Sigma Chemical, Burlington, MA, USA) at 100 µg/kg BW, and the other piglets were intraperitoneally injected with the same amount of sterile saline (0.9% NaCl). The selection of LPS dosage was based on previous studies on weaned piglets [25]. All the selected piglets were euthanized 4 h later, and plasma and liver samples were collected (Figure 1B).

Animals were purchased from a Langfang commercial farm and housed in a nursery room. During the experiment, the diet for the piglets was formulated meeting the National Research Council (2012) nutrient requirements (Appendix A: Table A1 and Table A2).

The experiment one and experiment two composition and nutrient levels of the basal diet are shown in Table A1. The basal diet did not contain any antibiotic growth promoters and the form of diet was mash. Piglets were accommodated in slatted floor pens (1.7 m × 1.5 m) with unrestricted access to feed and water. The room’s initial temperature was 28 °C and was gradually reduced to 26 °C. The room was lit naturally and artificially, with ventilation provided by speed-controlled fans. Each pen had two drinking fountains and an adjustable trough. Standard farm procedures were followed for disinfection and vaccination.

### 2.3. Sample Collection

Experiment one: Body weight (BW) was recorded individually at the beginning and the end of the trial. Any culling or mortality was recorded daily and feed consumption was corrected for accordingly. Growth performance was evaluated by calculating the average daily gain (ADG), average daily feed intake (ADFI), and gain-to-feed ratio (G:F) for each pen. To determine the incidence of diarrhea, fecal scores were monitored daily by visually appraising each subject using the following five-point fecal consistency scoring system: 1 = hard, dry pellet; 2 = firm, formed stool; 3 = soft, moist stool that retains its shape; 4 = soft, unformed stool; and 5 = watery liquid that can be poured. A liquid consistency (score 4–5) was considered indicative of diarrhea [26]. The incidence of diarrhea (%) was calculated as a percentage of the number of piglets with diarrhea divided by the total number of piglets in each treatment. At the end of the experiment (day 21), blood was taken from each piglet through the jugular vein to the heparin tube, left for 30 min, and centrifuged at 3000 rpm for 10 min. The plasma was separated and stored at −20 °C for analysis. 

Experiment two: At the end of the experiment (day 21), the plasma of the selected piglets after LPS challenge (4 h) was collected. The specific procedure was the same as in experiment one. After LPS challenge for 4 h, the piglets were stunned by a portable electrical stunner (the output voltage is 220 V) and bled quickly to be euthanized. Liver samples were collected and placed in cryogenic vials (Corning Incorporated, New York, NY, USA), frozen in liquid nitrogen, and stored at −80 °C for analysis.

### 2.4. Assay of Antioxidant Indices

The antioxidant indicators in both the plasma and liver, including the activities of SOD, catalase (CAT), and GSH-Px, as well as the concentrations of MDA, and the plasma growth hormone concentrations of insulin-like growth factor 1 (IGF-1), were determined using commercial assay kits as instructed (Enzyme-linked Biotechnology Co., Ltd., Shanghai, China). The concentrations of SOD, MDA, CAT, and GSH-Px were determined by micromethod according to the instructions. The absorbance was determined by microplate spectrophotometer (Bio Tek Instruments, Inc, Shanghai, China) at the appropriate wavelength, and the sample concentration was calculated. The superoxide anion (O^2−^) was generated via the reaction system of xanthine and xanthine oxidase. This anion can interact with WST-8 to yield a water-soluble dye, formazan, which exhibits absorption at 450 nm. The activity of SOD, which can eliminate O^2−^, thereby inhibiting the formation of formazan, can thus be measured. MDA reacts with thiobarbituric acid (TBA) to produce a red product that has a maximum absorption peak at 532 nm. The content of lipids containing peroxides can be estimated by colorimetry. Concurrently, the absorbance at 600 nm was measured, and the difference in absorbance at these two wavelengths was used to calculate the MDA content. CAT has the ability to decompose H_2_O_2_, which has a characteristic absorption peak at 240 nm. Consequently, the absorbance of the reaction solution at 240 nm decreased over time. The activity of CAT can be calculated based on the rate of change in absorbance. GSH-Px catalyzes the oxidation of glutathione (GSH) by H_2_O_2_ to produce glutathione disulfide (GSSG). Glutathione reductase (GR) then catalyzes the reduction of GSSG by nicotinamide adenine dinucleotide phosphate (NADPH) to regenerate GSH, oxidizing NADPH to NADP+ in the process. NADPH has a characteristic absorption peak at 340 nm, while NADP+ does not. Therefore, the activity of GSH-Px can be calculated by measuring the rate of decrease in absorption at 340 nm. The IGF-1 concentration was determined using an ELISA kit, samples were measured using microplates pre-coated with porcine IGF-1-trapping antibodies, and color was developed using TMB substrates through incubation and thorough cleaning. TMB turned blue under peroxidase and eventually yellow under acid. Finally, the absorbance was measured at a wavelength of 450 nm using an enzyme labeler to calculate the concentration of the sample. The protein concentration of the liver crude enzyme fluid was determined using the BCA protein quantitative kit, and the specific steps were carried out according to the instructions (Beijing Huaxing Bochuang gene Technology Co., Ltd., Beijing, China).

### 2.5. Real-Time Quantitative PCR Analysis (qPCR)

The hepatic RNA extraction and quantitative polymerase chain reaction (qPCR) procedure follows the method outlined by Cai et al. [8]. In succinct terms, for liver samples, the total RNA extraction employed Trizol reagent (Beijing AidLab Biotechnology Co., Ltd., Beijing, China) in accordance with the manufacturer’s stipulations. Firstly, we took 50 mg of liver tissue and added it to 1 mL of lysis buffer for tissue homogenization, then incubated it at room temperature for 5 min. Next, we added 0.2 mL of chloroform, vigorously shacked it for 15 s, and then incubated it at room temperature for 3 min. Afterwards, the sample was centrifuged at 12,000 rpm at 4 °C for 10 min using a centrifuge (Hitachi Koki Co., Ltd., Tokyo, Japan). Then, we took the supernatant, added anhydrous ethanol equal to half its volume, mixed well, and transferred it into an RA adsorption column, then centrifuged it at 12,000 rpm for 45 s. Subsequently, we added 500 μL of RE deproteinization solution, centrifuged it at 12,000 rpm for 45 s, and then discarded the waste liquid. Next, we added 500 μL of RW wash solution, centrifuged it at 12,000 rpm for 45 s, and repeated this step once. Then, we centrifuged the sample at 13,000 rpm for 2 min, added 60 μL of Rnase-free water, let it stand at room temperature for 2 min, centrifuged it at 12,000 rpm for 1 min, and finally obtained the liver RNA. The concentration and quality of RNA were scrutinized using a Nano Drop™ One/One Cmicro UV-Vis spectrophotometer (Thermo Fisher Scientific, Inc., Boston, MA, USA). The instrument calibration was performed using Rnase-free water prior to detection. Following this, complementary deoxyribonucleic acid (cDNA) was synthesized by a two-step reverse transcriptional procedure using the appropriate concentration of the reagent according to the quantitative concentration of liver RNA and the kit instructions (Beijing Takara Biomedical Technology Co., Ltd., Beijing, China). The cDNA was diluted with Rnase-free water at the appropriate concentration, packaged, and stored for further detection. The qPCR analysis was executed utilizing a CFX96 Touch real-time fluorescent qPCR system (Bio-Rad Laboratories Inc., Berkeley, CA, USA). The relative expression of the target gene was determined by employing the 2^−ΔΔCT^ method, wherein glyceraldehyde-3-phosphate dehydrogenase (GAPDH) functioned as the designated housekeeping gene. The specific primer sequences utilized in the qPCR assay can be found in Appendix B: Table A3.

### 2.6. Statistical Analysis

SPSS 19 (IBM, Armonk, NY, USA) was used for the statistical analysis. The univariate analysis of variance (ANOVA) and Tukey’s honest significance difference test were used for the statistical analysis. In addition, the Kruskal–Wallis test for non-normally distributed data sets was used to determine statistical significance. Orthogonal polynomial comparison tests with linear and quadratic effects were used to evaluate the effects of different doses of FC supplementation. Data are expressed as the mean of the standard error (SE). The difference was considered to be significant when *p* < 0.05, and the difference was considered to have a trend when 0.05 ≤ *p* < 0.10.

## 3. Results

### 3.1. Growth Performance and Diarrhea Incidence

The effects of different doses of FC supplementation on the growth performance of low-weaning weight piglets are presented in Figure 2A–E. Compared with the FC 0 group, dietary FC 150, FC 300, and FC 600 significantly increased the G:F ratio of low-weaning weight piglets (*p* < 0.05), and dietary FC 300 and FC 600 tended to increase the final BW (*p* = 0.059) and ADG (*p* = 0.057). There were no significant differences in the ADFI among experimental groups (*p* > 0.05). In addition, FC supplementation linearly increased the final BW, ADG, and G:F ratio (*p* < 0.05), and tended to linearly increase the ADFI (*p* = 0.097). In addition, there was a quadratic effect of FC supplementation on the G:F ratio (*p* < 0.05).

The result of different doses of FC supplementation on the diarrhea incidence of low-weaning weight piglets in experiment one is shown in Figure 2F. Compared with the FC 0 group, FC 300 supplementation significantly reduced the diarrhea incidence of low-weaning weight piglets from day 0 to 21 (*p* < 0.05). However, there was no significant difference in diarrhea incidence among FC 0, FC 150, and FC 600 groups (*p* > 0.05).

### 3.2. Plasma IGF-1 Concentrations

The effects of different doses of FC supplementation on the plasma IGF-1 concentrations of low-weaning weight piglets are shown in Figure 3E. Compared with the FC 0 group, the plasma IGF-1 concentrations in the FC 300 group were significantly increased (*p* < 0.05). In addition, the plasma IGF-1 concentrations linearly increased with the increase in FC supplemental concentrations (*p* < 0.05). There was no quadratic effect of FC supplementation on the plasma IGF-1 concentrations (*p* > 0.05).

### 3.3. Plasma Antioxidant Enzyme Activity

The effects of different doses of FC supplementation on the plasma antioxidant enzyme activity of low-weaning weight piglets are shown in Figure 3A–D. FC 300 and FC 600 supplementation significantly increased the plasma SOD activity and reduced the plasma MDA concentration compared to the FC 0 group (*p* < 0.05). There were no significant effects on plasma GSH-Px and CAT activities in all treatment groups (*p* > 0.05). In addition, FC supplementation linearly and quadratically increased the plasma SOD activity (*p* < 0.05) and linearly reduced the plasma MDA concentration (*p* < 0.05).

In addition, the effects of FC supplementation on plasma antioxidant enzyme activity of low-weaning weight piglets under LPS challenge are shown in Figure 4A–D. LPS challenge decreased the plasma SOD activity and increased the MDA concentration of low-weaning weight piglets (*p* < 0.05). Compared with the LPS group, the plasma SOD activity in the LPS + FC group was increased (*p* < 0.05), and the MDA concentration tended to be decreased in the LPS + FC group (*p* = 0.091). In addition, dietary FC significantly increased the plasma SOD activity and decreased the plasma MDA content compared to the CON group (*p* < 0.05).

### 3.4. Hepatic Antioxidant Enzyme Activity

The effects of FC supplementation on hepatic antioxidant enzyme activity of low-weaning weight piglets under the LPS challenge are shown in Figure 4E–H. Compared with the CON group, LPS reduced SOD activity and increased MDA concentration in the liver of low-weaning weight piglets (*p* < 0.05). Compared with the LPS group, the activity of CAT in the liver of LPS + FC piglets was significantly increased (*p* < 0.05), and the concentration of MDA was decreased in the LPS + FC group (*p* < 0.05), and dietary FC to LPS-challenged piglets tended to increase the activity of SOD (*p* = 0.087). In addition, hepatic CAT activity in the FC group was significantly increased compared with CON group (*p* < 0.05). 

### 3.5. Hepatic Antioxidant Genes mRNA Expression

The effects of FC supplementation on the hepatic antioxidant gene mRNA expression in low-weaning weight piglets under LPS challenge are shown in Figure 5A–I. Compared with the CON group, LPS challenge down-regulated *GCLC* mRNA expression in the liver of low-weaning weight piglets (*p* < 0.05). Compared with the LPS group, administering dietary FC to LPS-challenged piglets up-regulated *CAT* mRNA expression (*p* < 0.05) and down-regulated *Keap1* mRNA expression (*p* < 0.05) in the liver. The mRNA expression of *Nrf2* (*p* = 0.072), *SOD1* (*p* = 0.091), and *GCLM* (*p* = 0.094) in the LPS + FC group tended to be up-regulated compared to the LPS group. In addition, FC supplementation up-regulated the mRNA expressions of *Nrf2*, *SOD1*, and *GCLM* and down-regulated the mRNA expression of *Keap1* compared to the CON group (*p* < 0.05). In addition, all treatment groups had no significant effects on liver *HO1* and *NQO1* gene expression (*p* > 0.05).

### 3.6. Hepatic Mitochondrial Genes mRNA Expression

The effects of FC supplementation on hepatic mitochondrial gene mRNA expression in low-weaning weight piglets under LPS challenge are shown in Figure 6A–J. Compared with the CON group, LPS challenge down-regulated the expression of the mitochondrial fusion gene *MFN2* and division gene *FIS1* mRNA (*p* < 0.05) and tended to the decrease the division gene DRP1 (*p* = 0.051) in the liver of low-weaning weight piglets. Conversely, supplementation with FC partially improved mitochondrial biogenesis gene expression in the liver (Figure 6A–E). Compared with the LPS group, administering dietary FC to LPS-challenged piglets up-regulated the expression of the mitochondrial fusion gene *MFN2* mRNA in the liver (*p* < 0.05). In addition, the analysis of the expression of genes related to mitochondrial respiratory chain membrane proteins (Figure 6F–J) showed that supplementation with FC significantly up-regulated the expression of *SDHA* and *UQCRB* genes in the liver of weaned piglets stimulated by LPS (*p* < 0.05). The Spearman correlation analysis found significant correlations among *Keap1/Nrf2* signaling pathway genes, antioxidant enzyme-related genes, and mitochondrial respiratory chain membrane protein-related genes (Figure 6K).

## 4. Discussion

Weaning usually induces an increase of ROS in piglets, particularly in low-weaning weight piglets; consequently, weaning can disrupt the balance of oxidation reduction, reduce antioxidant capacity, cause oxidative stress and oxidative damage in the tissues and intestine, and potentially lead to diarrhea, inhibited growth, and even death [5,7,27,28,29]. It is well understood that piglets with a lower weaning weight are more susceptible to post-weaning challenges than pigs with a heavier weaning weight [5,30]. Therefore, piglets with low-weaning weight need antioxidants with nutritional regulation functions to improve the antioxidant capacity of weaned piglets, reduce diarrhea, and promote growth post weaning.

Our results in experiment one demonstrated that fucoidan (FC) supplementation could linearly improve the growth performance of low-weaning weight piglets and the optimal dose of FC was 300 mg/kg. The findings of this study were in accordance with previous research, which found that supplementations with FC could increase feed conversion and improve the growth performance of weaned piglets [31]. Moreover, the supplementation of various forms and doses of FC can enhance the growth rate and health status of young chicks [32], weaned kids [33], fish [34], and Penaeus monodon [35]. Previous studies demonstrated that FC could reduce the diarrhea incidence of weaned piglets [36,37]. In this study, FC supplementation at varying concentrations in low-weaning weight piglets revealed a significant reduction in the incidence of diarrhea at a concentration of 300 mg/kg. Walsh et al. [37] reported that adding 240 mg/kg FC to the diet of weaned piglets can reduce the incidence of diarrhea. On the other hand, Rattigan et al. [38] found that adding 250 mg/kg FC effectively improved the fecal consistency of weaned piglets. Based on these results, we speculated that the source and processing techniques of FC might determine its optimal dosage and method of use. These findings provide us with valuable references, helping us to better understand and utilize FC.

The free radical metabolism and antioxidant systems of piglets may undergo disruption after weaning, thereby instigating oxidative stress responses, which have the potential to interfere with the host’s antioxidant system and disrupt the cellular redox equilibrium [7,8,38,39]. The antioxidant defense system, which primarily comprises antioxidant enzymes such as SOD, CAT, and GSH-Px, along with other non-enzymatic antioxidants, can eliminate ROS [40,41]. MDA, the end product of lipid peroxidation, is frequently used as a marker for oxidative damage [42]. The role of functional nutritional supplements in improving the antioxidant capacity of piglets is widely recognized [43,44]; however, the effectiveness would vary due to differences in the type, source, and dosage of supplements. Our previous studies showed that supplementation with yeast hydrolysate from *Kluyveromyces fragilis* at 10 g/kg could significantly increase the activity of SOD and reduce the concentration of MDA in the plasma of weaned piglets, thereby enhancing the antioxidant capacity [7]. Furthermore, our studies revealed that a supplementation with 400 mg/kg of silybin could efficaciously mitigate the redox imbalance in weaned piglets and counteract the growth retardation induced by paraquat [8]. Our results in experiment one showed that supplementation with 300 and 600 mg/kg FC could enhance the activity of SOD and decrease the concentration of MDA in the plasma of piglets with a low-weaning weight. Consistent with previous studies, the use of FC or brown algae extracts to improve the antioxidant capacity of animals and inhibit oxidative stress has been verified in a variety of animal models [21,22,32,33,35]. Moreover, our study observed that the supplementation of FC at a dosage of 300 mg/kg notably elevated the concentrations of IGF-1 in the plasma of piglets with a low-weaning weight; the increase in IGF-1 concentrations exhibited a positive correlation with the piglets’ growth performance [45], thereby substantiating the growth-enhancing impact of FC supplementation. This is a finding that further corroborates the beneficial role of FC in promoting growth.

Although our preliminary investigation suggested that FC has the potential to augment the antioxidant and growth-enhancing capabilities of low-weaning weight piglets, this is far from sufficient to fully elucidate the mechanism of action of FC on oxidative stress in piglets. Therefore, we further designed an experiment and established an oxidative stress model using LPS-challenged low-weaning weight piglets to further study the alleviating effect of FC on oxidative damage. LPS has been substantiated as an effective inducer of oxidative stress in experimental animals, thereby validating its use in the establishment of oxidative stress models [25,46]. Our research findings indicate that LPS challenge triggers a pronounced systemic redox imbalance in low-weaning weight piglets, as evidenced by a decrease in SOD activity and an increase in MDA concentration in the plasma and liver. The challenge model in our study was similar to previous results, verifying the successful establishment of our experimental animal model [25,46,47]. The liver, being the primary organ for metabolism and detoxification, plays a crucial role in defending against severe infections and exogenous stimuli, and in tissue repair [48]. *Keap1* acts as a negative regulator of *Nrf2*, inhibiting its nuclear translocation. The *Nrf2* signaling pathway serves as the central regulator of oxidative stress responses within biological organisms, controlling the expression of various antioxidant response-related genes or enzymes [17]. When cells are damaged, *Nrf2* is up-regulated and activates phase II enzymes, enhancing the cells’ tolerance to oxidative stress [49]. Therefore, the *Keap1* regulation of *Nrf2* plays a crucial role in cellular resistance to oxidative stress and the maintenance of cellular homeostasis. In experiment two, our observations revealed that the supplementation with FC could down-regulate the expression of the *Keap1* gene, up-regulate the activation of the *Nrf2* pathway, and stimulate the expression of *CAT* antioxidant-related genes. This regulatory mechanism effectively mitigates oxidative stress, thereby offering substantial protection against liver damage induced by LPS to low-weaning weight piglets. This dual regulatory effect further underscores the potential of FC supplementation as a potent modulator of gene expression in the context of antioxidant defense mechanisms.

The mitochondrion, an essential organelle, is instrumental in controlling redox processes and lipid metabolism within liver cells [50]. Fission and fusion are crucial for maintaining mitochondrial balance by isolating and eliminating impaired components. When mitochondria malfunction, this may hinder fusion or stimulate fission to stop the integration of damaged parts into the healthy mitochondrial system [51]. The genes *FIS1* and *DRP1* play a regulatory role in mitochondrial division, while *MFN1*, *MFN2*, and *OPA1* primarily oversee the regulation of mitochondrial fusion [52]. Our research findings indicate that LPS challenges mitochondrial function by down-regulating the mRNA expression of *MFN2*, *FIS1*, and *DRP1*. Dietary supplementation with FC can partially reverse these adverse effects. Specifically, FC can alleviate LPS challenge-induced mitochondrial damage in the liver of low-weaning weight piglets by regulating the mitochondrial *MFN2* gene in the liver. Previous studies demonstrated that FC could prevent mitochondrial functional damage [22,53]. The mitochondrial oxidative phosphorylation system, which is crucial to cellular metabolism, consists of five enzyme complexes, including NADH dehydrogenase (Complex I), succinate dehydrogenase (Complex II), ubiquinol cytochrome c oxidoreductase (Complex III), cyanide sensitive oxidase (Complex IV), and ATP synthase (Complex V) [54,55]. Our findings demonstrated that the addition of FC counteracted the reduction in the expression of *SDHA* (Complex IV) and *UQCRB* (Complex II) genes in the liver of LPS-challenged low-weaning weight piglets. Thus, our results suggest that FC may enhance the activity of the mitochondrial oxidative phosphorylation system, preventing mitochondrial function damage.

## 5. Conclusions

The supplementation with an optimal dose of FC (300 mg/kg) exhibits benefits in the antioxidant capacity and growth performance of low-weaning weight piglets. The antioxidant function of FC might be attributed to the inhibited *Keap1* expression, controlled nuclear migration of *Nrf2*, enhanced CAT activity, various antioxidant enzymes, and improved mitochondrial function. Further, the antioxidant function may have eventually played a protective role against liver oxidative stress damage. Thus, the optimal dose of FC used in this study could provide a theoretical reference for the application of FC as a novel and natural antioxidant in swine production.

## Figures and Tables

**Figure 1 antioxidants-13-00407-f001:**
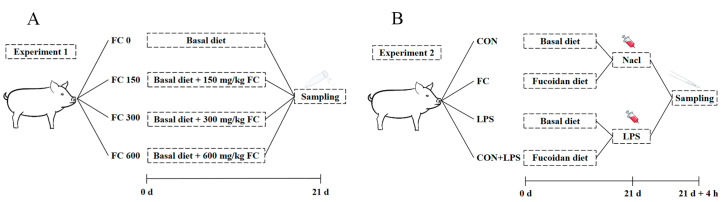
Schematic diagram of experimental design. (**A**) experiment one; (**B**) experiment two.

**Figure 2 antioxidants-13-00407-f002:**
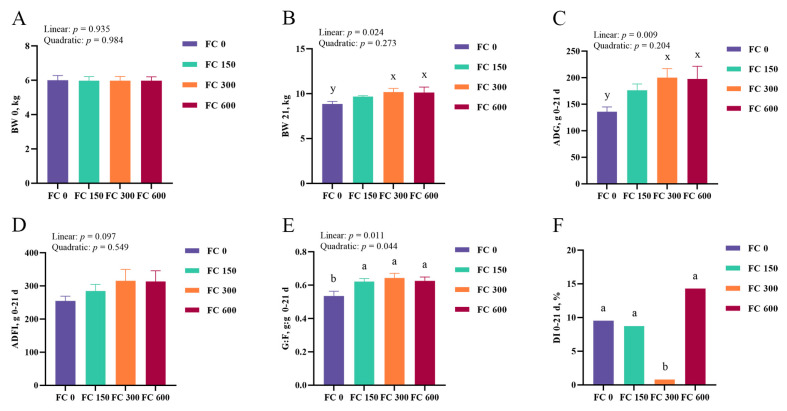
The effects of different doses of fucoidan (FC) supplementation on growth performance and diarrhea incidence of low-weaning weight piglets are presented at 0 to 21 days. (**A**,**B**) Body weight of piglets on day 0 (**A**) and 21 (**B**). (**C**–**E**) Average daily gain (**C**), average daily feed intake (**D**), and ratio of average daily gain to average daily feed intake (**E**) from 0 to 21 days. (**F**) Diarrhea incidence (DI) from day 0 to 21. Data were expressed as mean with their standard errors represented by vertical bars, (*n* = 6). Orthogonal polynomials were used to evaluate linear and quadratic responses to the concentrations of FC treatment. FC 0: basal diet, FC 150, FC 300, and FC 600 group, basal diet adding 150, 300, and 600 mg/kg FC, respectively. ^a,b^ Means listed in the same row with different superscripts are significantly different (*p* < 0.05). ^x,y^ Means listed in the same row with different superscripts showed a tendency to be different (0.05 ≤ *p* < 0.10).

**Figure 3 antioxidants-13-00407-f003:**
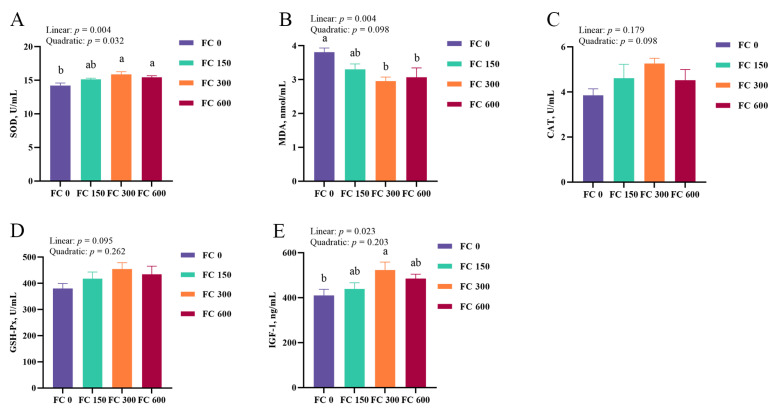
The effects of different doses of FC supplementation on plasma antioxidant enzyme activity and IGF-1 concentrations of low-weaning weight piglets is presented at day 21. (**A**–**D**) The plasma concentrations of indicators related to antioxidants (SOD, MDA, CAT, GSH-Px). (**E**) The plasma concentrations of IGF-1. Data were expressed as mean with their standard errors represented by vertical bars, (*n* = 6). Orthogonal polynomials were used to evaluate linear and quadratic responses to the concentrations of FC treatment. FC 0: basal diet, FC 150, FC 300, and FC 600 group, basal diet adding 150, 300, and 600 mg/kg FC, respectively. ^a,b^ Means listed in the same row with different superscripts are significantly different (*p* < 0.05).

**Figure 4 antioxidants-13-00407-f004:**
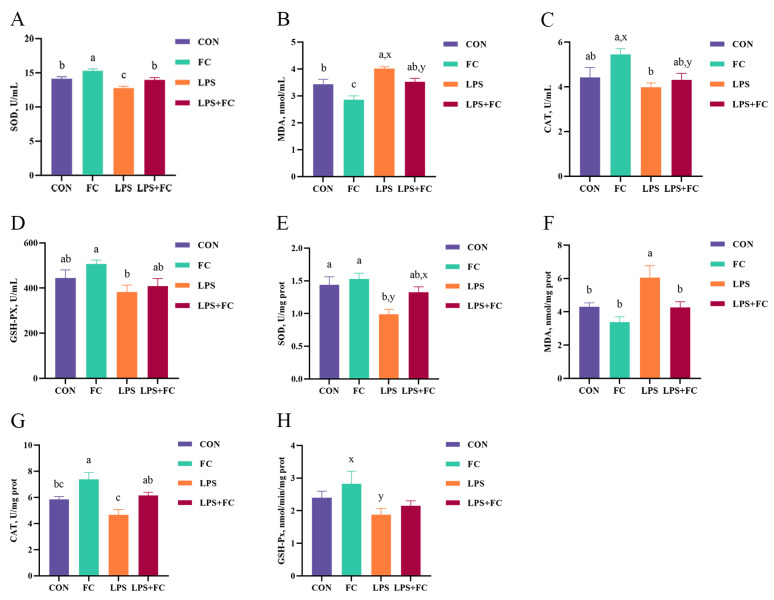
The effects of FC supplementation on the antioxidant enzyme activity of plasma and liver in LPS-challenged low-weaning weight piglets. (**A**–**D**) The plasma concentrations of indicators related to antioxidants (SOD, MDA, CAT, GSH-Px). (**E**–**H**) The liver concentrations of indicators related to antioxidants (SOD, MDA, CAT, GSH-Px). Data were expressed as mean with their standard errors represented by vertical bars, (*n* = 6). CON: basal diet, FC: basal diet + 300 mg/kg FC. ^a–c^ Means listed in the same row with different superscripts are significantly different (*p* < 0.05). ^x,y^ Means listed in the same row with different superscripts showed a tendency to be different (0.05 ≤ *p* < 0.10).

**Figure 5 antioxidants-13-00407-f005:**
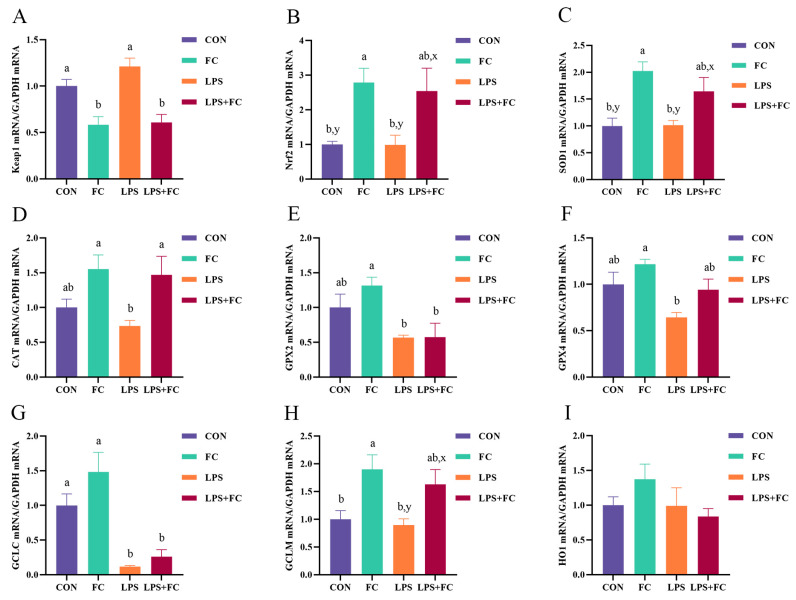
The effect of FC regulation and LPS challenge on liver antioxidant gene mRNA expression in low-weaning weight piglets. (**A**–**I**) Relative mRNA expression of *Keap1*, *Nrf2*, *SOD1*, *CAT*, *GPX2*, *GPX4*, *GCLC*, *GCLM*, and *HO-1*. Data were expressed as mean with their standard errors represented by vertical bars, (*n* = 6). CON: basal diet, FC: basal diet + 300 mg/kg FC. ^a,b^ Means listed in the same row with different superscripts are significantly different (*p* < 0.05). ^x,y^ Means listed in the same row with different superscripts showed a tendency to be different (0.05 ≤ *p* < 0.10).

**Figure 6 antioxidants-13-00407-f006:**
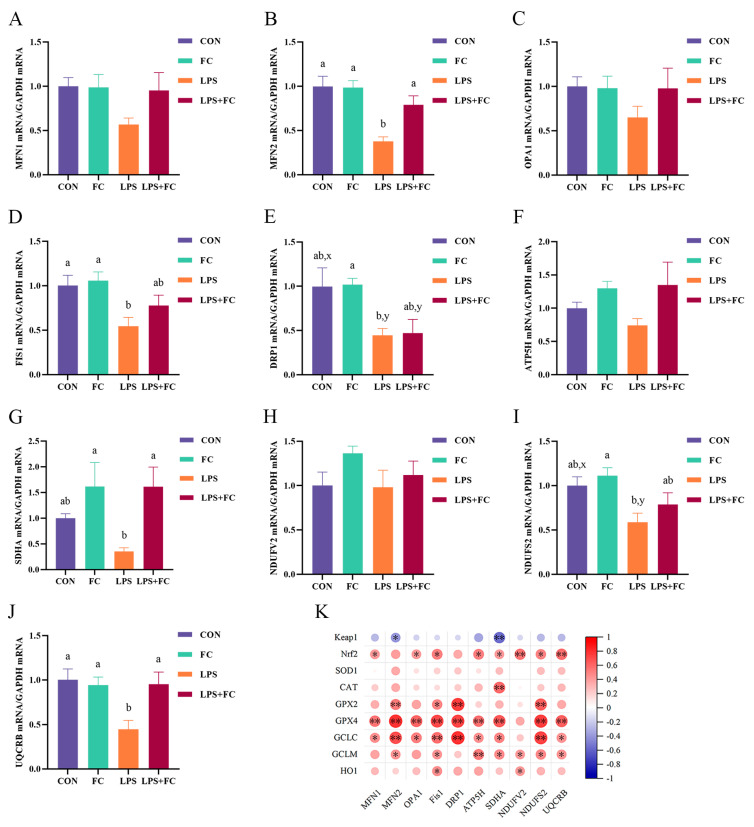
The effect of FC regulation and LPS challenge on liver mitochondrial biogenesis and respiratory chain membrane protein-related gene mRNA expression in low-weaning weight piglets. (**A**–**J**) Relative mRNA expression of *MFN1*, *MFN2*, *OPA1*, *FIS1*, *DRP1*, *ATP5H*, *SDHA*, *NDUFV2*, *NDUFS2*, and *UQCRB*. (**K**) The heatmap of Spearman’s correlation between the expression of mitochondrial function-related genes and *Keap1/Nrf2* signaling pathway genes. Data were expressed as mean with their standard errors represented by vertical bars, (*n* = 6). CON: basal diet, FC: basal diet + 300 mg/kg FC. ^a,b^ Means listed in the same row with different superscripts are significantly different (*p* < 0.05). ^x,y^ Means listed in the same row with different superscripts showed a tendency to be different (0.05 ≤ *p* < 0.10). * *p* < 0.05, ** *p* < 0.01. (**A**) MFN1 = mitofusin 1, (**B**) MFN2 = mitofusin 2, (**C**) OPA1 = optic atrophy 1, (**D**) FIS1 = mitochondrial fission protein 1, (**E**) DRP1 = dynamin-related protein 1, (**F**) ATP5H = ATP synthase, H+ transporting, mitochondrial F0 complex, subunit, (**G**) SDHA = succinate dehydrogenase complex flavoprotein subunit A, (**H**) NDUFV2 = NADH ubiquinone oxidoreductase core subunit V2, (**I**) NDUFS2 = NADH ubiquinone oxidoreductase core subunit S2, and (**J**) UQCRB = ubiquinol cytochrome c reductase binding protein. (**K**) Keap1 = kelch-like ECH-associated protein l, Nrf2 = nuclear factor-erythroid 2-related factor 2, SOD1 = superoxide dismutase 1, CAT = catalase, GPX = glutathione peroxidase, GCLC = glutamate-cysteine ligase catalytic subunit, GCLM = glutamate-cysteine ligase modifier subunit, HO1 = heme oxygenase 1.

## Data Availability

All data is included in the article.

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
