# Peer review of "Fucoidan Supplementation Improves Antioxidant Capacity via Regulating the Keap1/Nrf2 Signaling Pathway and Mitochondrial Function in Low-Weaning Weight Piglets"

_antioxidants, 2024, doi:10.3390/antiox13040407_

Round 1
Reviewer 1 Report
The relevance of the experiments is well supported by the details of the design and results. The results contribute to the knowledge on oxidative stress in a persistent problem (low weight weaned piglets) in commercial production. The research provides a good foundation to how FC supports improved performance in these piglets.
There are a range of minor grammatical issues that should be addressed by the authors. One common comment to be made is the need for authors to refrain from using 'levels' when actually the assays used are measuring concentrations.
Review: Fucoidan supplementation improves antioxidant capacity via 2 regulating the
Keap1/Nrf2 signaling pathway and mitochondrial function in low-weaning weight piglets
Abstract
The abstract is far too long. The journal direction is ‘The abstract should be a total of about 200 words maximum’
Ln 25: remove ‘a total of 24 piglets with 25 days 25 age low-weaning weight piglets’
Add ‘a total of 24 low-weaning weight piglets at 25 days of age’
Ln 37: remove ‘level’ add ‘concentration’
Introduction
Ln 52: remove ‘the modern’ add ‘current’
Ln 53: remove ‘helping to increase to enhance’ add ‘helping to enhance’
Ln 57; remove ‘weak’ add ‘poor’
Ln 61: remove ‘their’
Ln 77: change ‘[7, [8],’ to ‘[7, 8],
‘In the text, reference numbers should be placed in square brackets [ ], and placed before the punctuation; for example [1], [1–3] or [1,3].’ From journal instructions
Ln 79: remove ‘was’
Ln 82: ‘in vivo’ change to ‘in vivo’
Ln 86: remove ‘In another’ add ‘A further’
Ln 90: remove ‘increases’ add ‘increasing’ Also remove ‘capacity’
Ln 91 ‘tissue.’ Remove full stop
Ln 92: remove ‘prevents’ add ‘preventing’
Ln 96: remove ‘it needs’ add ‘there is a need’
Methods
Ln 172 & 173: remove ‘level’ add ‘concentrations’ Note is concentrations that are measured by the assay not levels.
Note This should be corrected throughout the manuscript
Ln 173; remove ‘was determined, were gauged with’
Add ‘was determined, were gauged with’ were determined using’
Ln 180: remove ‘, liver samples total RNA extraction employing’ add ‘, for liver samples the total RNA extraction employed’
Results
Ln 291 & 300 & 309 & 315 & 325: remove ‘tend tendency to be different (0.05 ≤ P < 0.10).’
Add ‘showed a tendency to be different (0.05 ≤ P < 0.10).’
Ln 217-223, 378, 379: remove ‘level’ add ‘concentrations’ Note is concentrations that are measured by the assay not levels.
Discussion
Ln 335: ‘[5, [7, [23, [24, [25]. [5, [26]’ Is this the correct formatting.
Ln 351: remove ‘of weaned’ add ‘to the diet of weaned’
Ln 365: remove ‘be’
Ln 381: remove ‘which finding further’ add ‘a finding that further’
Ln 388: remove ‘in LPS challenged low-weaning weight piglets’ it just repeats the same point made in Ln 387.
Ln 423: remove ‘Many previous’ add ‘previous’ avoid using adjectives such as many especially when the authors provide 2 references.
Ln 432: remove ‘prevent’ add preventing’
Ln 433: remove ‘excellent’ avoid using such adjectives in scientific manuscripts. How is it decided to be excellent? Allow the reader to decide the degree of excellence.
Author Response
The relevance of the experiments is well supported by the details of the design and results. The results contribute to the knowledge on oxidative stress in a persistent problem (low weight weaned piglets) in commercial production. The research provides a good foundation to how FC supports improved performance in these piglets.
There are a range of minor grammatical issues that should be addressed by the authors. One common comment to be made is the need for authors to refrain from using 'levels' when actually the assays used are measuring concentrations.
Response: Thank you very much for your comments. We have carefully revised the manuscript according to your comments.
Comments 1: The abstract is far too long. The journal direction is ‘The abstract should be a total of about 200 words maximum’
Response 1: Thank you for the reminder. We did our best to limit the length of the abstract following the journal direction, however, we could minimize the content to 237 words in order to keep the structural integrity of 2 experiments in the abstract. Please see Lines 19-35.
Comments 2: Ln 25: remove ‘a total of 24 piglets with 25 days 25 age low-weaning weight piglets’,Add ‘a total of 24 low-weaning weight piglets at 25 days of age’
Response 2: Thank you for your suggestion. This section has been changed to control the number of words in the abstract. Please see Line 22.
Comments 3: Ln 37: remove ‘level’ add ‘concentration’
Response 3: We have revised and flagged this content according to your suggestion. Please see Line 27.
Comments 4: Ln 52: remove ‘the modern’ add ‘current’
Response 4: We have revised the sentence according to your suggestion. Please see Line 41.
Comments 5: Ln 53: remove ‘helping to increase to enhance’ add ‘helping to enhance’
Response 5: We have revised the sentence according to your suggestion. Please see Line 42.
Comments 6: Ln 57; remove ‘weak’ add ‘poor’
Response 6: We have revised the word according to your suggestion. Please see Line 46.
Comments 7: Ln 61: remove ‘their’
Response 7: The word has been removed. Please see Line 48.
Comments 8: Ln 77: change ‘[7, [8],’ to ‘[7, 8],‘In the text, reference numbers should be placed in square brackets [ ], and placed before the punctuation; for example [1], [1–3] or [1,3].’ From journal instructions
Response 8: Thank you for pointing this out, we have corrected and marked all reference labels in the manuscript.
Comments 9: Ln 79: remove ‘was’
Response 9: The word has been removed.
Comments 10: Ln 82: ‘in vivo’ change to ‘in vivo’
Response 10: We have revised the type according to your suggestion. Please see Line 81.
Comments 11: Ln 86: remove ‘In another’ add ‘A further’
Response 11: We have revised the words according to your suggestion. Please see Line 86.
Comments 12: Ln 90: remove ‘increases’ add ‘increasing’ Also remove ‘capacity’
Response 12: We have revised the words according to your suggestion. Please see Line 89.
Comments 13: Ln 91 ‘tissue.’ Remove full stop
Response 13: The word has been removed.
Comments 14: Ln 92: remove ‘prevents’ add ‘preventing’
Response 14: We have revised the word according to your suggestion. Please see Line 91.
Comments 15: Ln 96: remove ‘it needs’ add ‘there is a need’
Response 15: We have revised the words according to your suggestion. Please see Line 96.
Comments 16: Ln 172 & 173: remove ‘level’ add ‘concentrations’ Note is concentrations that are measured by the assay not levels. Note This should be corrected throughout the manuscript
Response 16: Thank you for pointing this out. This problem has been corrected throughout the manuscript.
Comments 17: Ln 173; remove ‘was determined, were gauged with’, Add ‘was determined, were gauged with’ were determined using’
Response 17: We have revised the content according to your suggestion. Please see Lines 171-172.
Comments 18: Ln 180: remove ‘, liver samples total RNA extraction employing’ add ‘, for liver samples the total RNA extraction employed’
Response 18: We have revised the content according to your suggestion. Please see Lines 201-202.
Comments 19: Ln 291 & 300 & 309 & 315 & 325: remove ‘tend tendency to be different (0.05 ≤ P < 0.10).’ Add ‘showed a tendency to be different (0.05 ≤ P < 0.10).’
Response 19: We have revised the content according to your suggestion. Please see Lines 326 & 336 & 345 & 352 & 362.
Comments 20: Ln 217-223, 378, 379: remove ‘level’ add ‘concentrations’ Note is concentrations that are measured by the assay not levels.
Response 20: We have revised the word according to your suggestion. Please see Lines 252-258, 413, 414.
Comments 21: Ln 335: ‘[5, [7, [23, [24, [25]. [5, [26]’ Is this the correct formatting.
Response 21: Thank you for pointing this out. We agree that this is not the correct format and we have corrected the manuscript in full.
Comments 22: Ln 351: remove ‘of weaned’ add ‘to the diet of weaned’
Response 22: We have revised this content according to your suggestion. Please see Line 387.
Comments 23: Ln 365: remove ‘be’
Response 23: The word has been removed.
Comments 24: Ln 381: remove ‘which finding further’ add ‘a finding that further’
Response 24: We have revised this content according to your suggestion. Please see Lines 417-418.
Comments 25: Ln 388: remove ‘in LPS challenged low-weaning weight piglets’ it just repeats the same point made in Ln 387.
Response 25: The sentence has been removed.
Comments 26: Ln 423: remove ‘Many previous’ add ‘previous’ avoid using adjectives such as many especially when the authors provide 2 references.
Response 26: We have revised the words according to your suggestion. Please see Line 457.
Comments 27: Ln 432: remove ‘prevent’ add preventing’
Response 27: We have revised the word according to your suggestion. Please see Line 465.
Comments 28: Ln 433: remove ‘excellent’ avoid using such adjectives in scientific manuscripts. How is it decided to be excellent? Allow the reader to decide the degree of excellence.
Response 28: We have removed the word according to your suggestion.
Reviewer 2 Report
Dear authors, congratulations for this interesting and well written article, that in my opinion deserves to be published in Antioxidants.
Minor Comments:
The study showed impressive results and my comments are summed up as follows :
1- The keap1Nrf2 pathway's role in improving antioxidants has to be clarified in the introduction, this is very important also considering the aim of the journal
Author Response
Dear authors, congratulations for this interesting and well written article, that in my opinion deserves to be published in Antioxidants.
Response: Thank you very much for your encouragement, and we modified the manuscript according to your comments.
Comments 1: The keap1Nrf2 pathway's role in improving antioxidants has to be clarified in the introduction, this is very important also considering the aim of the journal
Response 1: Thank you for pointing this out. We added the role of keap1/Nrf2 signaling pathway in antioxidant in the introduction of this paper. Please see Lines 66-77.
Comments 2: Material and method parts 2.4 and 2.5 are poorly defined and lacking details, need to be extended and more clarified in all parts
Response 2: We supplemented and marked the operational details in parts 2.4 and 2.5 of Materials and methods in the manuscript. Thank you for your suggestion. Please see Lines 173-195, 203-222.
Comments 3: Check if the in-text citation style has to be according to the journal rule
Response 3: Thank you for pointing this out. We have double checked and revised and marked the manuscript in full in accordance with the journal rules.